# Combined Surgical and Orthodontic Treatments in Children with OSA: A Systematic Review

**DOI:** 10.3390/jcm9082387

**Published:** 2020-07-26

**Authors:** Laura Templier, Cecilia Rossi, Manuel Miguez, Javier De la Cruz Pérez, Adrián Curto, Alberto Albaladejo, Manuel Lagravère Vich

**Affiliations:** 1Division of Orthodontics, School of Dentistry, University of Alfonso X el Sabio, 28016 Madrid, Spain; Faculty of Medicine; lauratemplier@hotmail.fr (L.T.); cecilia.rossi.uni@hotmail.it (C.R.); jdela@uax.es (J.D.l.C.P.); 2Faculty of Medicine, University of Salamanca, 37007 Salamanca, Spain; adrian_odonto@usal.es (A.C.); albertoalbaladejo@hotmail.com (A.A.); 3Sleep Dental Medicine Spanish Society (SEMDeS), Dental Sleep Medicine Program, Catholic University of Murcia UCAM, 30107 Murcia, Spain; miguez@infomed.es; 4Division of Orthodontics, School of Dentistry, Faculty of Medicine and Dentistry, University of Alberta, Edmonton, AB T6G 2R3, Canada

**Keywords:** surgical, orthodontic treatments, apnea

## Abstract

Obstructive sleep apnea (OSA) is a sleeping breathing disorder. In children, adenotonsillar hypertrophy remains the main anatomical risk factor of OSA. The aim of this study was to assess the current scientific data and to systematically summarize the evidence for the efficiency of adenotonsillectomy (AT) and orthodontic treatment (i.e., rapid maxillary expansion (RME) and mandibular advancement (MA)) in the treatment of pediatric OSA. A literature search was conducted in several databases, including PubMed, Embase, Medline, Cochrane and LILACS up to 5th April 2020. The initial search yielded 509 articles, with 10 articles being identified as eligible after screening. AT and orthodontic treatment were more effective together than separately to cure OSA in pediatric patients. There was a greater decrease in apnea hypoapnea index (AHI) and respiratory disturbance index (RDI), and a major increase in the lowest oxygen saturation and the oxygen desaturation index (ODI) after undergoing both treatments. Nevertheless, the reappearance of OSA could occur several years after reporting adequate treatment. In order to avoid recurrence, myofunctional therapy (MT) could be recommended as a follow-up. However, further studies with good clinical evidence are required to confirm this finding.

## 1. Introduction

Obstructive sleep apnea (OSA) is described as a sleeping breathing disorder, characterized by prolonged partial upper airway obstruction and/or intermittent complete obstruction [1]. This syndrome is commonly correlated with intermittent hypoxemia and sleep fragmentation [2]. The prevalence of OSA has been estimated, in a general orthodontic population, by questionnaires and it was found to be 10.8%, which is more than double that reported by similar methods in a healthy pediatric population [3].

OSA has also been associated with frequent snoring, disturbed sleep, daytime neurobehavioral problems, neurocognitive impairments, academic underperformance, hypertension, cardiac dysfunction and systemic inflammation. Daytime sleepiness may occur but is uncommon in young children [4]. Etiological factors include any condition that reduces the caliber of the upper airways, such as craniofacial dysmorphism, hypertrophy of lymphoid tissues, obesity, hypotonic neuromuscular diseases and neuromotor control alterations during sleep. However, adenotonsillar hypertrophy remains the main anatomical risk factor [4,5,6,7].

Therefore, adenotonsillectomy (AT) is the recommended first-line treatment for pediatric OSA in children with adenotonsillar hypertrophy [4,8,9,10]. It has been demonstrated that AT reduced the severity of OSA in most children, and reduced symptoms and improved behavior, quality of life and polysomnographic findings [9]. However, a significant number of patients with pediatric OSA undergoing AT exhibit residual persistent post-surgery OSA [10].

Moreover, it was proven that children with OSA and large tonsils had some craniofacial morphology characteristics like a narrow and long face, a narrow upper airway, maxillary constriction and/or some degree of mandibular retrusion [11,12,13,14,15,16]. Hence, AT was not always successful in controlling OSA in children, and orthodontic treatments such as rapid maxillary expansion (RME) or mandibular advancement (MA) could be a helpful complement. Nowadays, there are a lot of systematic reviews and meta-analyses about OSA treatments but none has compared the different treatments together and the information about both treatments in concurrence is very limited.

The aim of this systematic review was to assess the current scientific data and to summarize, in a systematic manner, evidence for the efficiency of a combination of surgery (e.g., AT) and orthodontic treatment (i.e., RME and MA) in the treatment of pediatric OSA.

## 2. Materials and Methods

A Preferred Reporting Items for Reporting Systematic reviews and Meta-Analyses (PRISMA) protocol was followed for reporting this systematic review [17].

### 2.1. Protocol and Registration

Protocol registration was not available.

### 2.2. Eligibility Criteria

The Population, Intervention, Comparison, Outcomes and Study design (PICOS) process was used to select abstracts and potential articles retrieved from the databases. The inclusion criteria were:Population: children diagnosed with OSA by polysomnography (PSG) or by a home sleep study.Intervention: subjects who underwent surgery such as AT and orthodontic treatment (i.e., RME, MA). RME and MA were searched for individually since the focus was on orthodontic treatment (either RME or MA or both together with surgery (tonsillectomy or adenoidectomy)).Comparison: a combination of clinical assessments to evaluate the efficiency of surgery and orthodontic treatment to resolve OSA.Outcomes: three main outcomes were evaluated: severity of OSA, oxygen saturation and recurrence of OSA after treatment.Study design: randomized, non-randomized trials, cohort and case-control studies, case series and case reports were included.

The exclusion criteria were syndromic patients, animal studies, book or conference abstracts, systematic reviews and meta-analyses. Studies comparing AT and expansion independently as treatments were also excluded. There were no restrictions on language, year or status of publication for inclusion.

### 2.3. Information Sources

A literature search was conducted online in several databases, including PubMed, Embase, Medline, Cochrane and LILACS up to 5 April 2020.

### 2.4. Search Strategy

The search was performed using keywords, combinations of keywords with truncations, medical subject headings (Mesh) and Boolean logical operators such as “OR” to be more sensitive. The search strategy is presented in Table A1.

Additional potentially relevant articles were identified by performing a manual search via Google, looking for reference lists of retrieved articles.

### 2.5. Study Selection

The selection of the studies consisted of two phases. During the first phase, two reviewers (L.T. and C.R.) independently identified and checked the titles and abstracts of all records. Those references that met the eligibility criteria were included. Full texts of references containing insufficient information in the title and/or abstract for a decision on inclusion or exclusion were retrieved for evaluation in phase two. In the second phase of article selection, the same two reviewers evaluated the full texts of the remaining articles. Those studies that met the eligibility criteria were included. In cases of disagreement, in both phases, a third reviewer (M.L.V.) settled by consensus.

### 2.6. Data Collection Process

Two authors (L.T. and C.R.) independently extracted and reviewed data from the included studies. Any disagreement was discussed between them.

### 2.7. Data Items

From the included studies, various data were collected, such as authors, year, sample size, age, gender, body mass index (BMI), types of screening used to diagnose OSA and types of treatments. They are summarized in Table 1.

### 2.8. Risk of Bias in Individual Studies

To assess the methodological quality/risk of bias in trials and case–control studies, the first and second authors independently used the checklist by Downs and Black [28], consisting of 26 items categorized in five subgroups: Reporting (nine items), External validity (three items), Bias (seven items), Confounding (six items) and Power (one item). For each item, one point was scored when the respective question was answered “yes” except as described in the original paper for question 5 (Reporting subscale) which can be scored 0, 1 or 2 and question 27 (Power subscale) which can be scored 0, 1, 2, 3, 4 or 5. However, as a study either has or does not have sufficient power to detect a clinically important effect, question 27 was scored, in the present study, 0 or 1. A score of 26 to 28 was considered excellent, 20 to 25 good, 15 to 19 fair and 14 or below was considered to have a poor clinical importance effect. The risk of bias in individual studies is shown in Table A2.

Moreover, the same authors evaluated case reports with the CAse REport (CARE) checklist. It is composed of 13 items: Title, Keywords, Abstract, Introduction, Patient information, Clinical findings, Timeline, Diagnostic assessments, Therapeutic intervention, Follow-up and outcomes, Discussion, Patient perspective and Informed consent. Items are divided by subscale. Each question is answered with “yes” or “no” [29]. To evaluate the different case reports, we gave 1 for the answer: “yes” and 0 for “no” and made the sum to compare them. The accuracy and transparence of the case reports are reported in Table A3.

### 2.9. Summary Measures

The main outcomes assessed were: apnea hypoapnea index (AHI), respiratory disturbance index (RDI), mean of the lowest oxygen saturation, nadir oxygen saturation, average oxygen saturation and the ODI (oxygen desaturation index) at different times: before both treatments (initial), after the first treatment of surgery or orthodontic treatment (intermediate) and after both treatments (final). These outcomes are presented in Table 2 and Table 3.

## 3. Results

### 3.1. Selection of Studies

The information flow of the search and selection of studies is shown in Figure 1. Following the electronic database searches, 505 articles were identified and screened for retrieval and four additional records were identified through other sources. Among the initially identified articles, 259 studies were retrieved after the removal of duplicates. Thus, in the first selection phase, a total of 244 articles were excluded on the basis of title and abstract. In the second phase, on the examination of their full texts, five articles were eliminated and the reasons for exclusion were: overview article (*n* = 2), not related to OSA (*n* = 1), patients who did not receive both treatments, i.e., surgical and orthodontic treatment (*n* = 1) and retracted article (*n* = 1). Therefore, ten studies met all the inclusion criteria and remained for quantitative synthesis.

### 3.2. Study Characteristics

The included studies were categorized into one randomized controlled trial (RCT) [27], two non-randomized controlled trials (NRCTs) [22,26], two observational case–control studies [24,25] and five case reports [18,19,20,21,23]. The sample sizes ranged from one to 80 subjects. The mean age of participants before starting treatment ranged from 3 to 12 years. In three case reports [19,23,25], the participants were female. In the other studies, males made up a higher proportion than females. The mean BMI varied between 15.75 and 22.4, however, only five studies reported the BMI [19,22,23,25,26]. To diagnose and evaluate the severity of OSA, eight studies [19,20,21,22,23,24,25,27] employed PSG and two [18,26] used home sleep studies (HSTs). All study participants received surgery and orthodontic treatment. Among the surgical treatments, AT was the most commonly used, but there was also one case report [21] which performed other types of surgery: epiglottoplasty and a reduction of the tongue base. Two types of orthodontic treatment were carried out: RME or MA. However, only two case reports used MA as an orthodontic treatment [19,20]. Concerning the order of treatment, five studies [19,20,23,24,25] performed AT before orthodontic treatment, one case report [21] performed both treatments at the same time, one study [18] completed the orthodontic treatment before AT and three trials [22,26,27] compared both in different groups: AT followed by RME and RME followed by AT.

### 3.3. Risk of Bias Within Studies

The level of evidence in the trials and case–control studies was assessed by the Downs and Black checklist questionnaire. Two studies [22,26] had a low level of evidence and were evaluated as poor. One case–control study was qualified as fair [24]. An RCT trial [27] and a case-control study [25] were estimated as good. The main reasons for downgrading the quality of evidence pertained to the inclusion of case reports and non-randomized studies with critical methodological issues that most probably introduced bias. Villa et al. [22] compared three groups: one group was treated by AT, the second one by RME and the last one was treated by both. In comparison to the other two groups, the third one had a small numbers of subjects. Therefore, there was a large age difference between each group. The mean age for groups 1, 2 and 3 were 3.7 ± 0.92, 6.58 ± 1.83 and 4.6 ± 3.2, respectively. In the same way, in the trial of Pirelli et al. [26], there were discrepancies among the distribution of characteristics of patients in each group: subjects with indicators of chronic adenotonsillar inflammatory problems were placed in the group to be initially treated with AT, while those not clearly presenting this problem were placed in the initial orthodontic treatment group.

As case reports are considered weak evidence in the hierarchy of research evidence, all of them were classified as having a low level of evidence. However, to evaluate their accuracy and transparency, the CARE checklist was applied. Among the five case reports, Gracco et al.’s had the best rating [21]. The worst score was given to Nauert et al.’s case report [20].

### 3.4. Results of Individual Studies

In these studies, three main outcomes were assessed: the severity of OSA, oxygen saturation and the recurrence of OSA after surgical and orthodontic treatment.

#### 3.4.1. Severity of OSA

The severity of OSA was evaluated by different measures, such as AHI or RDI. The AHI is the number of apneas or hypopneas recorded during the study per hour of sleep, whereas the RDI means the average number of episodes of apnea, hypopnea and respiratory event-related arousals per hour of sleep. Unlike the AHI, the RDI counts not only respiratory events during sleep, but it also takes into consideration respiratory effort-related arousals which can be defined as arousals from sleep [30]. 

To evaluate the effectiveness of treatment, most of the studies reported and compared the initial and final AHI. Eight papers [18,19,20,23,24,25,26,27] also described the intermediate AHI, that is, after patients underwent the first treatment. Only three studies [23,25,27] assessed the RDI.

All studies reported a higher decrease in the AHI or RDI after both treatments (surgery and orthodontic treatment) [18,19,20,21,22,23,24,25,26,27]. According to Guilleminault et al. [27], there was no significant difference between the group beginning with orthodontic treatment and the one beginning with surgical treatment after the first phase of treatment. The means of the intermediate AHI were 5.4 ± 0.6 and 4.9 ± 0.6, respectively.

On the other hand, Pirelli et al. [26] reported a greater effectiveness of RME as an initial treatment parameter. In the RME group, 15 subjects (37.5%) had a normal clinical evaluation and a normal polygraphy at the initial post-treatment evaluation, four months after the completion of treatment; 17 presented a significant improvement (AHI 6.5 ± 3.1) and eight had minimal or no improvement (AHI 13 ± 3.5). However, in the AT group, only six patients (15%) presented total remission, 18 presented an improvement in OSA (AHI 6 ± 3.1) and 16 had minimal or no improvement (AHI 15 ± 2.9).

In the same way, two case reports [20,23] appeared not to respond to AT and showed an improvement in the symptoms of pediatric OSA after undergoing orthodontic treatment. Kim et al. [23] described a pre-RME AHI, final AHI and a two-and-a-half-year follow-up AHI of 18.9, 4.4 and 1, respectively.

Furthermore, Villa et al. [22] evaluated and compared the persistence of OSA in children who only underwent AT or RME and in children who received both. They described how in approximately 40% of the children who underwent the surgical procedure, there was a complete resolution of OSA. One year after treatment, subjects who underwent RME treatment were found to have a higher post-treatment AHI than those who underwent AT even though they had a mild form of the disease prior to treatment. The one-year post-RME AHI and the one-year post-surgery AHI were 2.64 ± 3.11 and 1.79 ± 1.82, respectively, and these results were significant. In the group treated by AT and RME, there was a significant decrease in AHI from the beginning (AHI initial: 10.14 ± 7.25) to one year after the end of treatment (AHI final: 0.88 ± 0.95).

#### 3.4.2. Oxygen Saturation

To measure the oxygen saturation, various outcomes were assessed: the mean of the lowest oxygen saturation, the nadir oxygen saturation, the average oxygen saturation and the ODI.

Oxygen saturation is the fraction of oxygen-saturated hemoglobin relative to total hemoglobin (unsaturated + saturated) in the blood. Many patients suffering from OSA have intermittent oxygen desaturation associated with periods of apnea or hypopnea [31]. The nadir oxygen saturation may refer to the lowest point of oxygen saturation, whereas the ODI is the number of times that the blood oxygen level drops by a certain degree from the baseline per hour of sleep.

Three trials [24,25,27] measured the mean of the lowest oxygen saturation before, after the first phase of treatment and at the end of both treatments. All these experimental studies found a significant increase in the lowest oxygen saturation, and it was higher after both treatments (surgery and orthodontic treatment) but not after the first phase of treatment. 

Two case reports [19,23] described the nadir oxygen saturation. Kim et al. [23] reported an increase from 60% to 94% of the nadir oxygen saturation compared to before RME and two and a half years after orthodontic treatment, in a child who did not respond to AT. Bignotti et al. [19] illustrated an increase in nadir oxygen saturation after AT from 89% to 93% but a decrease from 93% to 50% after twin block treatment.

Three articles [19,21,22] reported the average of oxygen saturation. Bignotti et al. [19] have shown a lower level of oxygen saturation in patients who have undergone both treatments (AT and twin block), as opposed to Gracco et al. [21], who described a higher level of oxygen saturation after surgery and RME. Similarly, the third study [22] reported a significant increment in the average oxygen saturation one year after surgical treatment or RME therapy. However, they did not find any differences in mean overnight oxygen saturation in the group treated by RME and AT before and one year after treatment (97.85 ± 1.28% vs. 97.42 ± 2.06%). 

ODI was assessed by two case reports at the beginning, after surgery treatment and at the end of both treatments. In these articles, ODI decreased by a higher amount after both treatments than after only surgical treatment [19,21].

#### 3.4.3. Recurrence

Two case–control studies [24,25] had a large follow-up of their patients and reported the recurrence of OSA in patients who were treated by AT followed by RME. Guilleminault et al. [25] evaluated and monitored a group of 29 teenagers considered cured of OSA (AHI 0.4 ± 0.4) and with no clinical complaints after undergoing an AT and RME in early childhood. After several years, 20 of the 29 subjects presented with clinical complaints and a mean AHI of 3.1 ± 1.0, whereas nine patients did not report clinical complaints and had a mean AHI of 0.5 ± 0.2. Likewise, Guilleminault et al. [24] evaluated 24 subjects treated with AT followed by RME, with or without follow-up myofunctional re-education. Thirteen of the 24 subjects who did not undergo myofunctional re-education developed the recurrence of symptoms with a mean AHI of 5.3 ± 1.5 and a mean minimum oxygen saturation of 91 ± 1.8%. All the 11 subjects who completed myofunctional re-education for 24 months revealed healthy results (AHI 0.5 ± 0.4). 

## 4. Discussion

### 4.1. Summary of Evidence

#### 4.1.1. Severity of OSA

All the included studies [18,19,20,21,22,23,24,25,26,27] highlighted a major decrease in polysomnographic indexes (AHI or RDI) after undergoing surgical and orthodontic treatments. Most children needed both treatments to have complete resolution of their symptoms and a normalization of PSG [22,27]. 

These results were independent of the different types of treatment used. However, most studies performed the same types of treatment: RME as an orthodontic treatment and AT as a surgical treatment. Only two case reports [19,20] performed MA as an orthodontic treatment and one case report [21] described an epiglottoplasty and a reduction of the tongue base as surgical treatments. Therefore, the type of surgical treatment should be determined depending on the obstruction site. Hence, in patients with residual OSA after undergoing AT and RME, additional sites of obstruction during sleep could be considered, such as epiglottis collapse [32], supraglottic collapse or tongue base collapse [33].

Besides, except in two trials classified as having a high risk of bias [22,26], there were no differences observed between the various first treatment approaches, that is to say, between subjects who began with orthodontic treatment or with surgical treatment. Pirelli et al. [26] reported a greater effectiveness of RME as an initial treatment parameter, however, it was not a randomized trial and the selection of patients which had AT was not adequate. Conversely, Villa et al. [22] highlighted a higher post-treatment AHI in the group who underwent RME than those who underwent AT. However, in this study, there was an important difference between the mean age in the group treated by AT and the group treated by RME.

#### 4.1.2. Oxygen Saturation

Two trials with a low risk of bias and one with a moderate risk of bias found a significant increase in the mean of the lowest oxygen saturation, and it was higher after receiving both surgery and orthodontic treatment than after the first phase of treatment [24,25,27]. Concerning the nadir oxygen saturation, the results were heterogeneous and had a high risk of bias [19,23].

Among the studies which evaluated the average oxygen saturation, there were a lot of discrepancies between the results and a lack of strong evidence. After undergoing both treatments, the mean level of oxygen saturation was lower according to Bignotti et al. [19], higher as reported by Gracco et al. [21] and did not change according to Villa et al. [22].

In the same way, two case reports of poor clinical relevance described a higher ODI reduction in patients who were subjected to both treatments than those undergoing only surgical treatment [19,21].

#### 4.1.3. Recurrence

The recurrence of OSA after AT and RME were reported in two retrospective case–control studies [24,25]. In the first one [25], characterized to be of good clinical relevance, Guilleminault et al. highlighted a reoccurrence of OSA in 20 of the 29 patients treated in their childhood by AT and RME. Thus, they suggested that the reappearance of OSA could occur several years after reporting an adequate treatment, following adequate surgical and orthodontic treatment.

Interestingly, they assessed that 12 of the 20 teenagers with sleep-related complaints had the same Friedman scale score of 4, and 16 of the 20 children with OSA recurrence had “high and narrow hard palates” and 14 of the 20 children had “an overjet of more than 2.5 mm”, suggesting that these patients presented skeletal relapse despite prior maxillary expansion.

In the same way, the second case–control study [24], which related the recurrence of OSA, was evaluated as having a moderate risk of bias. It showed the follow-up of 24 children with (*n* = 11) or without MT (*n* = 13). All the children were cured (AHI 0.4 ± 0.3) by the combination of AT and palatal expansion.

The children who received MT over the long term remained cured of OSA compared to children who were never trained to perform these exercises and they subsequently had a recurrence of OSA. Similar results were reported by Villa et al. [34] in post-adenotonsillectomy patients who were randomized to either receive MT or not.

Thus, as various studies reported the benefits of the combination of orthodontic or surgical treatment with myofunctional re-education on breathing, speech, swallowing, orofacial growth and the elimination of abnormal head–neck posture, MT could be considered effective as a follow-up therapy to avoid the recurrence of OSA in children treated by adenotonsillectomy and orthodontic treatment.

### 4.2. Importance of Pediatric Treatment

Sleep disorders in children occur during the critical period of brain development. The consequences of not treating them can be of high relevance, leading to the following health conditions: stunted growth, cognitive and behavioral abnormalities such as hyperactivity, poor school performance, cardiovascular and endothelial dysfunction and an overall reduced quality of life. That is why it is deemed important to treat pediatric patients with OSA [9,35].

#### 4.2.1. Multidisciplinary Approaches

As OSA is a multifactorial syndrome [36], a multidisciplinary approach should be taken to treat OSA in children. That is why, when combined soft tissue surgery, orthodontic treatment and myofunctional therapy worked more effectively together, reducing the AHI [18,19,20,21,22,23,24,25,26,27], which was irrespective of the order in which the treatments were performed [27].

In the same way, AT and RME treatments affect the growth patterns of patients with OSA in a positive way. One non-randomized trial assessed the craniofacial changes after AT and RME in mouth-breathing children. Nevertheless, this article was excluded from our quantitative analysis because it did not tie together AT and RME with OSA. This study compared children with oral breathing treated (*n* = 33) or not (*n* = 20) by AT. In the group of children subjected to AT, 17 of the 33 underwent RME. The authors found that AT controlled the facial vertical growth but not maxillomandibular sagittal growth. However, in children subjected to surgery and RME, they noticed that the vertical growth pattern was controlled, and the maxillomandibular sagittal measurements were significantly changed, with a consequent improvement in facial profile. Furthermore, in the frontal view, the group treated by AT and RME showed a significant cross-sectional gain in maxillary width and nasal width measures [37].

#### 4.2.2. Optimal Age

Any child aged 1 to 18 years old could be a candidate for tonsillectomy [38]. The most common late complications of AT were dehydration or secondary post-tonsillectomy hemorrhage (PTH) [39]. In the retrospective study of Lindquist et al. [40], 5225 patients under the age of eighteen years were identified, with an overall late complication rate of 12.8%. Patients younger than three years of age were more likely to present dehydration. This was most significant for children under 2 years of age. However, PTH was more common in older children. 

RME has to be performed before the fusion of maxillary sutures, which is completed at the age of 14–15 in females and 15–16 in males [41]. According to Melsen et al. [42], in the early stage (up to 10 years old), the suture was smooth and broad. In the juvenile stage (from 10 to 13 years old), it started to have overlapping sections. Finally, during the adolescent stage (13 and 14 years of age), the suture was wavier with increased interdigitations. In patients with an advanced stage of skeletal maturation, orthopedic maxillary expansion was not possible.

Hence, for the treatment of skeletal class II malocclusion with functional appliance, it has been shown that the functional treatment was efficient when it was performed during the pubertal growth spurt [43,44,45]. However, if it is performed before the pubertal growth spurt, class II functional appliance will not have clinically relevant effects to correct the skeletal relationship. Nonetheless, there was a dentoalveolar correction, effective in reducing overjet and severity of malocclusion [46]. In one case report [20], after an AT treatment failure, a patient of three years old received a functional therapy of class II. However, in early cases with class II malocclusion it is recommended that only the transversal deficiency of the maxilla is treated [47].

### 4.3. Limitations

At the systematic methodological review level, no reportable limitations exist, as the PRISMA guidelines were followed, and two reviewers independently selected articles, extracted data and evaluated the clinical relevance to reduce selection bias.

At the study level, the most important limitation was that most of the articles retrieved displayed limited to poor clinical evidence and this was the reason why it was not possible to assess a meta-analysis.

One notable weakness that impacted the methodological quality/risk of retrieved articles was that, in most of studies, the number of subjects undergoing surgical and orthodontic treatment was too low.

One of the limitations in our review was that, in most studies, treatments were only applied in young children. Before undergoing treatment, only two case reports [19,23] had patients older than ten years and the oldest participant was twelve years old [19]. 

Another important limitation was that various studies did not report the BMI of their population. However, OSA syndrome is considered as one of the adverse consequences of childhood obesity. Narang et al. [48] reported that OSA occurred in up to 60% of obese children. In the same way, Mitchell et al. [49] showed that obese children are more likely to have a higher level of pre- and post-adenotonsillectomy OSA when compared with children of normal weight.

## 5. Conclusions

A limitation present in this review was the availability of few studies and most of them were considered to have a high risk of bias. Nevertheless, considering the available information, AT and orthodontic treatment were more effective together rather than separately to cure OSA in pediatric patients. There was a greater decrease in AHI and RDI, a major increase in the mean of the lowest oxygen saturation and the ODI in patients after undergoing both treatments. The reappearance of OSA could occur several years after reporting adequate treatment. In order to avoid recurrence, MT could be recommended as a follow-up. Further research with good clinical evidence is required to confirm this finding.

## Figures and Tables

**Figure 1 jcm-09-02387-f001:**
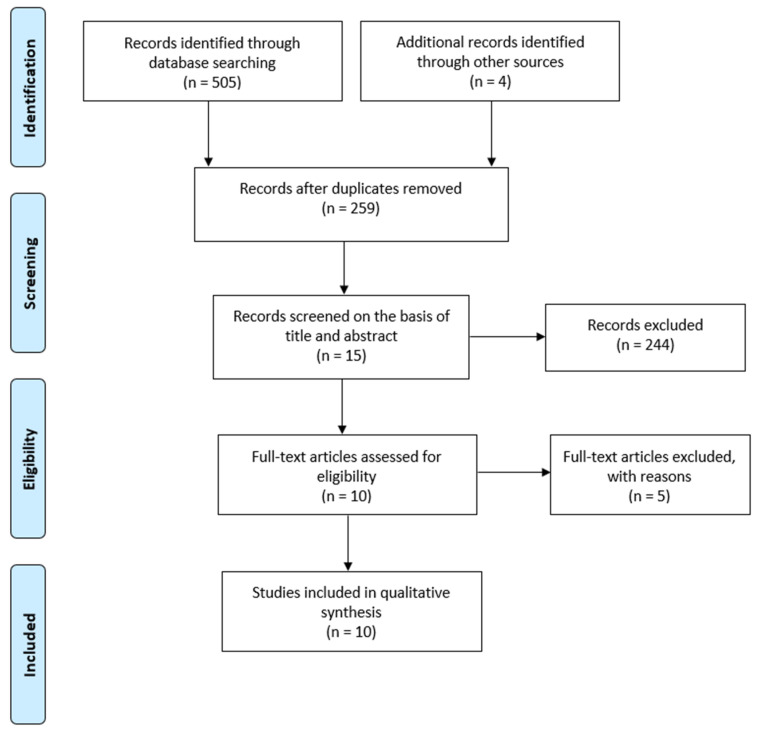
Flow chart of selection process.

**Table 1 jcm-09-02387-t001:** Study characteristics.

Year—Principal Author	Type of Study	Type of Treatment	Type of Screening	Sample Size	Age of Participants (Year) Mean + SD	Sex	BMI (kg/m^2^) Mean + SD
**2019 Alexander et al. [18] **	Case report	RME followed by AT	HST	2	9	F	/
**2019 Bignotti et al. [19] **	Case report	AT followed by twin block	PSG	1	12	M	22.2
**2019 Nauert [20]**	Case report	AT followed by Bionator	PSG	1	3	F	/
**2018 Gracco et al. [21] **	Case report	At the same time: RME + epiglottoplasty + reduction of the tongue base	PSG	1	8	F	/
**2014 Villa et al. [22] **	NRCT	Group 1: AT: 25 Group 2: RME: 22 Group 3: AT + RME: 5	PSG	52	Group 1: 3.7 ± 0.92 * Group 2: 6.58 ± 1.83 * Group 3: 4.6 ± 3.2	Group 1 and 2: 34M/13F Group 3: 3M/2F	Group 1: 15.75 ± 1.82 * Group 2: 18.82 ± 3.44 * Group 3: 16.65 ± 3.65
**2014 Kim [23]**	Case report	AT followed by RME F: Final treatment FU: Follow-up 2–5 years after treatment	PSG	1	11	M	22.4
**2013 Guilleminault et al. [24]**	Case—Control	AT followed by RME, Follow-up: MT or WMT	PSG	24 ^†^: Group MT: 11 Group WMT: 13	I: 5.5 ± 1.2 F: 7.3 ± 1.5 FU: 11.6 ± 1.2	14M/10F	/
**2013 Guilleminault et al. [25] **	Case—Control	Follow-up study of OSA in teenagers after AT + RME treated in their childhood	PSG	29	I: 7.6 ± 1.7 F: 8.6 ± 2.8 FU: 14.4 ± 0.9	20M/9F	NR: 15.9 ± 1.9 R: 15.7 ± 2.1
**2012 Pirelli et al. [26] **	NRCT	Group 1: RME: 40 Group 2: AT: 40 Group 3: Residual OSA: RME + AT and AT + RME: 42	HST	Group 1 and 2: 80 Group 3: 42	7.3	43M/37F	<24
**2011 Guilleminault et al. [27] **	RCT	Group 1: AT followed by RME, Group 2: RME followed by AT	PSG	31: Group 1: 16 Group 2: 15 ^†^	6.5 ± 0.2	14M/17F	/

RCT—randomized controlled trial; NRCT—non-randomized controlled trial; OSA—obstructive sleep apnea; AT—adenotonsillectomy; RME—rapid maxillary expansion; MT—myofunctional therapy; WMT—without myofunctional therapy; PSG—polysomnography; HST—home sleep study; I—before treatment; F—final treatment; FU—follow-up; R—patients with relapse; NR—patients without relapse; M—male; F—female; * *p* < 0.05; ^†^ One patient did not have AT.

**Table 2 jcm-09-02387-t002:** Summary of severity of OSA.

Year—Principal Author	Type of Treatment	AHI Initial (Events/h) Mean + SD	AHI Intermediate (Events/h) Mean + SD	AHI Final (Events/h) Mean + SD	RDI Initial (Events/h) Mean + SD	RDI Intermediate (Events/h) Mean + SD	RDI Final (Events/h) Mean + SD
**2019 Alexander et al. [18] **	RME followed by AT	Patient A: 74 Patient B: 16	Post RME: Patient A: 11 Patient B: 4	Patient A: 0.9 Patient B: 1.6	/	/	/
**2019 Bignotti et al. [19] **	AT followed by twin block	25.5	Post AT: 3.4	0.7	/	/	/
**2019 Nauert [20]**	AT followed by Bionator	/	Post AT: 10.2	5-year follow-up: normal cognitive development and any evidence of OSA	/	/	/
**2018 Gracco et al. [21] **	At the same time: RME + epiglottoplasty + reduction of the tongue base	21.8	/	0.6	/	/	/
**2014 Villa et al. [22] **	Group 1: AT: 25 Group 2: RME: 22 Group 3: AT + RME: 5	Group 1: 17.25 ± 13.94 * Group 2: 5.81 ± 6.05 * Group 3: 10.14 ± 7.25	/	Group 1: 1.79 ± 1.82 * Group 2: 2.64 ± 3.11 * Group 3: 0.88 ± 0.95	/	/	/
**2014 Kim et al. [23] **	AT followed by RME F: Final treatment FU: Follow-up 2–5 years after treatment	/	18.9	F: 4.4 FU: 1	/	19.8	F and FU: 5.9
**2013 Guilleminault et al. [24] **	AT followed by RME, Follow-up: MT or WMT	10.5 ± 2.6	Post AT^†^: 4.3 ± 1.6	F: 0.4 ± 0.3 MT: 0.5 ± 0.4 * WMT: 5.3 ± 1.5	/	/	/
**2013 Guilleminault et al. [25] **	Follow-up study of OSA in teenagers after AT + RME treated in their childhood	9 ± 5	Post AT: 3 ± 4	F: 0.4 ± 0.4 NR: 0.5 ± 0.2 * R: 3.1 ± 1 *	15 ± 6.4	Post AT: 7 ± 6	F: 0.6 ± 0.5 NR: 1.5 ± 1.2 * R: 7 ± 1.2
**2012 Pirelli et al. [26]**	Group 1: RME: 40; Group 2: AT: 40; Group 3: Residual OSA: RME + AT and AT + RME: 42	Group 1 and 2: 12.8	Group 3: RME + AT: 13 ± 3.5 AT + RME: 15 ± 2.9	Group 1 (6/40) and G2 (15/40): 6.5 ±3.1 Group 3: 39/42 patients were cured	/	/	/
**2011 Guilleminault et al. [27] **	Group 1: AT followed by RME, Group 2: RME followed by AT	Group 1: 12.5 ± 0.8 Group 2: 11.1 ± 0.7	Group 1: 4.9 ± 0.6 Group 2: 5.4 ± 0.6	Group 1: 0.9 ± 0.3 Group 2: 0.9 ± 0.3	Group 1: 21.3 ± 1.0 Group 2: 19.5 ± 1.0	Group 1: 8.0 ± 0.7 Group 2: 7.9 ± 0.5	Group 1: 1.6 ± 0.6 Group 2: 1.7 ± 0.8

OSA—obstructive sleep apnea; AT—adenotonsillectomy; RME—rapid maxillary expansion; MT—myofunctional therapy; WMT—without myofunctional therapy; F—final treatment; FU—follow-up; R—patients with relapse; NR—patients without relapse; * *p* < 0.05; ^†^ One patient did not have AT.

**Table 3 jcm-09-02387-t003:** Summary of oxygen saturation.

Year—Principal Author	Lowest SaO_2_ Initial (%) Mean + SD	Lowest SaO_2_ Intermediate (%) Mean + SD	Lowest SaO_2_ Final (%) Mean + SD	Average Sa0_2_ Initial (%) Mean + SD	Average SaO_2_ Intermediate (%) Mean + SD	Average SaO_2_ Final (%) Mean + SD	ODI Initial (Events/Hour)	ODI Intermediate (Events/h)	ODI Final (Events/h)
**2019 Alexander et al. [18] **	/	/	/	/	/	/	/	/	/
**2019 Bignotti et al. [19] **	Nadir: 89	Nadir: 93	Nadir: 50	97.3	96.0	96.0	22.0	0.7	3.2
**2019 Nauert [20] **	/	/	/	/	/	/	/	/	/
**2018 Gracco et al. [21] **	/	/	/	96.5%	/	98.1	23.4	/	1
**2014 Villa et al. [22] **	/	/	/	Group 1: 96.11 ± 2.7 * Group 2: 96.56 ± 1.47 * Group 3: 97.85± 1.28	/	Group 1: 97.50 ± 1.14 * Group 2: 97.42 ± 1.84 * Group 3: 97.42 ± 2.06	/	/	/
**2014 Kim e al. [23] **	/	Nadir: 60	Nadir FT: 85 Nadir: FU: 94	/	/	/	/	/	/
**2013 Guilleminault et al. [24] **	90 ±1.5	Post AT^†^: 92 ± 1	F: 95 ± 1 MT: 96 ± 1 * WMT: 91 ± 1.8	/	/	/	/	/	/
**2013 Guilleminault et al. [25] **	91 ± 2.5	Post AT: 94 ± 3	F: 98 ± 1.5 NR: 97 ± 1 * R: 92.5 ± 1.5 *	/	/	/	/	/	/
**2012 Pirelli et al. [26] **	/	/	/	/	/	/	/	/	/
**2011 Guilleminault et al. [27] **	Group 1: 92.1 ± 0.5 Group 2: 92.5 ± 0.4	Group 1: 95.2 ± 0.3 Group 2: 95.9 ± 0.3	Group 1: 98.0 ± 0.2 * Group 2: 97.6 ± 0.3 *	/	/	/	/	/	/

AT—adenotonsillectomy; MT—myofunctional therapy; WMT—without myofunctional therapy; F—final treatment; FU—follow-up; R—patients with relapse; NR—patients without relapse; SaO_2_—oxygen saturation; ODI—oxygen desaturation index; * *p* < 0.05; ^†^ One patient did not have AT.

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
