# Peer review of "Combined Surgical and Orthodontic Treatments in Children with OSA: A Systematic Review"

_jcm, 2020, doi:10.3390/jcm9082387_

Round 1

Reviewer 1 Report

I would like to thank the authors for the opportunity of reviewing this work. The topic of the work is surely interesting and not particularly investigated in international literature. 

The present version of the article can be improved and I would consider it for publication after some revisions. The main concern is that the discussion section is a repetition of the result section. The authors should try to discuss the results and not only repeat them.

For example, what could you expect compared to a non-treatment patient, how can the treatment strategy and time influence the results, the advantages of a multidisciplinary combined approach with respect to an only orthodontic or only surgical approach. 

Furthermore:

  • Moderate English editing is required to improve paper readability 
  • appendix 2 part 1 lacks the total score and grade

Author Response

Reviewer 1

I would like to thank the authors for the opportunity of reviewing this work. The topic of the work is surely interesting and not particularly investigated in international literature. 

Thanks for the positive comment.

The present version of the article can be improved and I would consider it for publication after some revisions. The main concern is that the discussion section is a repetition of the result section. The authors should try to discuss the results and not only repeat them. For example, what could you expect compared to a non-treatment patient, how can the treatment strategy and time influence the results, the advantages of a multidisciplinary combined approach with respect to an only orthodontic or only surgical approach. 

Thanks for the comment; we have added more information in the discussion section specially strengthening the points that seemed weak and thus seemed repetition of the results. Page 12-14

We deleted a sentence (page 12, paragraph 4.1.1) because of the addition of a new paragraph (page 13 and 14) “Importance of pediatric treatment” where we explained the importance and benefit of treatment for obstructive sleep apnea in pediatric patients and thus we divided it in two subsections:

  • 2.1 Multidisciplinary approaches to highlight the importance to combine treatments
  • 2.2. Optimal age where we explain if it exists an optimal age to perform each treatment. Page 14

Furthermore:

  • Moderate English editing is required to improve paper readability 
    • English has been reviewed by a English first language professional
  • appendix 2 part 1 lacks the total score and grade
    • We have added the scores and grades as suggested in the appendix

Reviewer 2 Report

It is a sensitive topic to perform orthognathic surgery in children. I think that must be stress if there are some negative influences in face bone development or not (it was mention that the if growing doesn’t conserve the initial result, at least keep the main pattern of development).

In spite of scarcity of publication in this field, the review was well done.

Author Response

It is a sensitive topic to perform orthognathic surgery in children. I think that must be stress if there are some negative influences in face bone development or not (it was mention that the if growing doesn’t conserve the initial result, at least keep the main pattern of development).

Thanks for the review, in the manuscript, when we referred to surgery we were meaning mostly soft tissue surgery not orthognathic surgery. We agree with the reviewer in terms that orthognathic surgery at that young on an age could have detrimental effects in normal growth and development of the child. We did add a comment in the discussion in terms that adenotonsillectomy which could improve the facial growth pattern of patients. Page 13

In spite of scarcity of publication in this field, the review was well done.

Thanks for the positive comment.

Reviewer 3 Report

The review had the objective to assess the current scientific data and to summarize in a systematic manner evidence of the efficiency from a combination including surgery (e.g.: AT) and orthodontic treatment (i.e.: RME and MA) in the treatment of pediatric OSA. The study has some problems that should be fixed before further analysis.

An error in bias study analysis was performed and out 10 studies 5 are case report. Retrieve substantial and effective data from such small samples could be ineffective 

Why the review was not registered 

Intervention. Please clarify if RME and MA are associate therapy or they have been searched as independent therapy.

Risk of bias: different risk of bias should be used for case report/serie and RCTs. The authors described they use Downs and Black checklist. This checklist s for randomized and not clinical study. For case report they should use https://www.care-statement.org.

Conclusion should underline the strong limits of the present data.

Author Response

Dear reviewer,

Thank you for your corrections. We have taken into account your comments and we answered and corrected as followed:

The review had the objective to assess the current scientific data and to summarize in a systematic manner evidence of the efficiency from a combination including surgery (e.g.: AT) and orthodontic treatment (i.e.: RME and MA) in the treatment of pediatric OSA. The study has some problems that should be fixed before further analysis.

Thanks for the feedback and we hope we have addressed your concerns in an acceptable way.

An error in bias study analysis was performed and out 10 studies 5 are case report. Retrieve substantial and effective data from such small samples could be ineffective 

We do agree with the reviewer, and taking that into consideration we have changed the way of evaluating the case reports as suggested.

Why the review was not registered 

We did indeed try to register the review using PROSPERO but unfortunately never got a confirmation nor email back, thus we placed it as not available.

Intervention. Please clarify if RME and MA are associate therapy or they have been searched as independent therapy.

They have been searched individually since the focus was for orthodontic treatment either RME, MA or both together with surgery (tonsillectomy or adenoidectomy).

Risk of bias: different risk of bias should be used for case report/serie and RCTs. The authors described they use Downs and Black checklist. This checklist s for randomized and not clinical study. For case report they should use https://www.care-statement.org.

We changed the section of “Risk of bias in individual studies” ( page 5) and the section of “Risk of bias within studies”(page 9). We added the Appendix 3 to evaluate the accuracy of the case-reports with the CARE  checklist.

Conclusion should underline the strong limits of the present data.

We have added limitation to the conclusion in relation to the information found.

Sincerely

Round 2

Reviewer 3 Report

Authors fulfilled reviewer comments and suggestions .

I rather prefer to  read at the beginning of the conclusions the importance of the limits of the study.

Author Response

Thanks for the review, we have moved the limitations at the start of the conclusion as recommended.

Thanks again